# Symptoms, Treatment, and Outcomes of COVID-19 Patients Coinfected with *Clostridioides difficile*: Single-Center Study from NE Romania during the COVID-19 Pandemic

**DOI:** 10.3390/antibiotics12071091

**Published:** 2023-06-22

**Authors:** Lidia Oana Stămăteanu, Ionela Larisa Miftode, Claudia Elena Pleșca, Olivia Simona Dorneanu, Manuel Florin Roșu, Ioana Diandra Miftode, Maria Obreja, Egidia Gabriela Miftode

**Affiliations:** 1Department of Internal Medicine II, Faculty of Medicine, University of Medicine and Pharmacy Gr. T. Popa, 700115 Iași, Romania; lidia-oana.stamateanu@umfiasi.ro (L.O.S.); claudia-elena.badarau@umfiasi.ro (C.E.P.); maria.n.obreja@d.umfiasi.ro (M.O.); egidia.miftode@umfiasi.ro (E.G.M.); 2“St. Parascheva” Clinical Hospital of Infectious Diseases, 700116 Iași, Romania; olivia.dorneanu@umfiasi.ro (O.S.D.); florin.rosu@umfiasi.ro (M.F.R.); 3Department of Preventive Medicine and Interdisciplinarity, Faculty of Medicine, University of Medicine and Pharmacy Gr. T. Popa, 700115 Iași, Romania; 4Department of Intensive Care Unit, Infectious Diseases Clinical Hospital, 700115 Iași, Romania; 5Department of Radiology, “St. Spiridon” Emergency Clinical Hospital, 700111 Iași, Romania; miftode.ioana-diandra@email.umfiasi.ro

**Keywords:** *Clostridioides difficile* infection, SARS-CoV-2, co-infections, antimicrobial resistance, antibiotic stewardship

## Abstract

The Coronavirus disease 2019 (COVID-19) pandemic has brought new challenges across medical disciplines, particularly in infectious disease medicine. In Romania, the incidence of SARS-CoV-2 (Severe acute respiratory syndrome coronavirus 2) infection increased dramatically since March 2020 until March 2022. Antibiotic administration for pulmonary superinfections in COVID-19 intensified and, consequently, increased rates of *Clostridioides difficile* infection (CDI) were hypothesized. We conducted a single-center, retrospective, observational study on patients from North-Eastern Romania to assess clinical characteristics and outcomes of COVID-19 and *Clostridioides difficile* (CD) coinfection, and to identify risk factors for CDI in COVID-19 patients. The study enrolled eighty-six CDI and COVID-19 coinfected patients admitted during March 2020–February 2021 (mean age 59.14 years, 53.49% men, 67.44% urban residents) and a group of eighty-six COVID-19 patients. On admission, symptoms were more severe in mono-infected patients, while coinfected patients associated a more intense acute inflammatory syndrome. The main risk factors for severe COVID-19 were smoking, diabetes mellitus, and antibiotic administration. Third generation cephalosporins (55%) and carbapenems (24%) were the main antibiotics used, and carbapenems were significantly associated with severe COVID-19 in patients coinfected with CD during hospitalization. Coinfection resulted in longer hospitalization and poorer outcomes. The extensive use of antibiotics in COVID-19, particularly carbapenems, contributed substantially to CD coinfection.

## 1. Introduction

The Severe Acute Respiratory Syndrome Coronavirus-2 (SARS-CoV-2) infection is currently the most combated infectious disease [1]. More than three years into the COVID-19 pandemic, the challenges posed by this pathogen are still remarkable. The main organ affected by the SARS-CoV-2 infection is the lung, and most patients experience specific respiratory symptoms. However, a significant number of patients have also reported gastrointestinal symptoms such as diarrhea, abdominal pain, nausea, and vomiting [2]. It is also worth mentioning that 2–33% of SARS-CoV-2 infected patients only presented diarrhea and nausea as initial symptoms [3,4].

*Clostridioides difficile* (CD) is an anaerobic, toxin-producing, gram-positive, spore-forming bacterium. It is a multidrug-resistant pathogen known as the principal cause of healthcare-associated diarrhea [5]. Noteworthy, *Clostridioides difficile* infection (CDI) is also considered a major public health challenge, associated with high recurrence risk, morbidity, and mortality [6].

The most significant risk factors for CDI are antibiotic use, old age, and hospitalization. The antibiotics most commonly associated with CDI are penicillin, cephalosporins, fluoroquinolones, and clindamycin. Patients over 65 years old are 8 to 10 times more likely to develop CDI than patients under 65 [7]. Clinical manifestations of CDI can range from asymptomatic carriage to watery stools, abdominal pain, nausea, vomiting, fever, loss of appetite, or physical asthenia [7]. A common symptom in both COVID-19 and CDI is diarrhea, making it difficult to pose the diagnosis of co-infection when only one of these two infections is initially suspected [8].

Over time, microorganisms have developed various mechanisms of resistance to antibiotics, which makes it crucial to administer the right antibiotic in the right dose, at the right time, and with the right duration. We now understand the way in which poor anti-microbial stewardship contributes to increased resistance, therefore limiting therapeutic options [9]. In the COVID-19 era, especially, the use of antibiotics has been disturbingly high even though antibiotics have no recognized benefits in the treatment of SARS-CoV2 infection without bacterial co-infection [10]. The use of broad-spectrum antibiotics in patients with COVID-19 modifies the function and causes an imbalance of intestinal microbiota, raising the susceptibility of being colonized by opportunistic pathogens, with CDI as one likely consequence [11].

During the pandemic, the prevalence of antibiotic usage among COVID-19 patients varied from approximately 63.1% in Europe, to as high as 87.5% in other areas of East–Southeast Asia. The elevated antibiotic consumption is not only associated to the occurrence of CDI, but the incidence of CDI is also influenced by the treatment duration and the specific type of antibiotics administered. Despite the significant role of antibiotic usage in the occurrence of CDI, an interesting observation was made during the pandemic. The epidemiology of CDI experienced modifications during this time period. Despite the sustained or even increased antibiotic consumption, the implementation of restrictions and heightened focus on hygiene practices, including strict adherence to protective measures by healthcare professionals, have contributed to a decrease in the number of CDI cases in some geographical regions, therefore a lower occurrence of nosocomial transmission. This underscores the significance of adhering to stringent hygiene measures and prioritizing disease prevention, not only during times of pandemics but as an ongoing practice in healthcare settings [12,13,14,15,16].

In addition, the immunosuppressive state encountered in SARS-CoV-2 infection predisposes to a variety of bacterial superinfections [17]. The treatment for the associated bacterial infections, especially locally or in South-Eastern Europe where antibiotic resistance rates are high, often consists of multiple antibiotic regimens, thus contributing to the risk of developing CDI [18,19,20].

## 2. Results

### 2.1. Baseline Patient Characteristics and Hospitalization Data

Eighty-six patients with confirmed SARS-CoV-2 infection and CDI coinfection as Group A and an equal number of patients with COVID-19 as Group B were included in this study from a total of 5176 patients with COVID-19 admitted in our hospital between March 2020 and January 2021 (1.66%).

Among the 86 patients with CDI and COVID-19, 31.2% (27) patients had received antibiotics prior to their hospitalization. Among those 27 patients, only 8 had experienced diarrhea before being admitted. The remaining patients received antibiotics during their hospital stay and subsequently developed gastrointestinal symptoms, prompting their attending physicians to conduct CDI testing.

In Group A, we performed a comparison between symptoms in patients with severe forms of COVID-19 (SpO2 ≤ 93% on room air, *N =* 36) and those with mild/moderate forms of COVID-19 (SpO2 > 94% on room air, *N =* 50), trying to assess the impact on the severity of COVID-19 symptoms. The statistical results show that, in coinfected patients, there were significant differences regarding the presence of dry cough between severe and mild/moderate cases (*p* = 0.047) (Table 1). A similar comparison based on the patients’ sex revealed that men presented significantly more dry cough (*p* = 0.044) and asthenia (*p* = 0.043) compared to women (Appendix A).

Further, we verified the effect of the independent sex-linked variable on oxygen saturation (OS) with the T test for independent samples. Interestingly, men had significantly lower SpO2 than women: t (84) = −2.073, *p* = 0.041, *p* < 0.05.

The statistical analysis of laboratory findings contributed to the identification of further significant differences between coinfected patients with different forms of COVID-19 (Table 2). Apart from the elevated leukocytes levels, lactate dehydrogenase (LDH), C-reactive protein (CRP), and erythrocyte sedimentation rate (ESR) in severe forms of coinfection, we noticed significant associations with lymphopenia (*p* = 0.03) and hyperglycaemia (*p* = 0.05).

The most commonly used antibiotics in the hospital, in the analyzed period, were 3rd generation cephalosporins (55%) and carbapenems (24%). Other antibiotics used were aminopenicillins (7%) and, to a considerably lesser extent, macrolides. In our study, following the diagnosis of a bacterial superinfection, 31.2% (27) patients received antibiotics prior to their hospitalization and the rest of the patients were administered antibiotics during hospitalization. In Group A, carbapenems were preferred in patients with severe forms of COVID-19 (*p* = 0.04), while cephalosporines were given mostly to patients with moderate COVID-19 (*p* = 0.009) (Table 3). The administration of antibiotics exhibited variability in terms of timing, with certain patients receiving them upon admission under the decision of medical practitioners, while others were prescribed antibiotics in the moment when clinical and paraclinical indicators suggestive of a bacterial superinfection appeared.

In Group B, specifically 34.8% (30) of the patients exhibited severe forms of COVID-19, while the remaining 56 (65.2%) individuals presented with mild/moderate forms. Concerning the administration of carbapenems in Group B, among the patients with severe forms, 11 individuals received carbapenem treatment. Conversely, among the patients with mild/moderate forms, only 4 individuals were prescribed carbapenems.

The presence of diabetes mellitus (DM) among coinfected patients was significantly associated with a severe form of COVID-19 (*p* = 0.01) (Appendix A).

Multivariate analyses were performed to identify predictors of ground-glass opacity visible on the computed tomography (CT)-scans of patients’ lung. In the initial regression model, dyspnea, fever, chest pain, DM, obesity, and cardiac pathologies associated had significant predictive value for ground-glass opacities pneumonia (*p* = 0.001) (Figure 1).

### 2.2. Analysis of Clinical and Paraclinical Features of CDI in Group A Patients

The results obtained in terms of symptomatology of COVID-19 in mono-infected and coinfected patients were significantly different: dyspnea (50% vs. 32%), asthenia (81.39% vs. 57%), anosmia (21% vs. 10%), and ageusia (19.76% vs. 9%). Contrary to initial expectations, the analysis revealed a lower incidence of COVID-19 symptoms in patients belonging to Group A compared to those in Group B.

The laboratory markers were examined in both groups, and we found that the higher levels of white blood cell (WBC) were significantly associated with coinfection status (*p* = 0.03).

CDI testing revealed that 80 cases (93%) tested positive for toxin A, 72 (83%) tested positive for toxin B, and 70 (81%) tested positive for both toxins. Additionally, the GDH antigen was identified in all patients. Testing for CDI upon hospitalization was conducted on a total of 17 patients (19.7%), while the remaining patients (80%) underwent testing for CDI during their hospitalization period. Out of the 17 patients tested upon admission, 8.1% (7) patients had a history of antibiotic use prior to hospitalization, while the remaining 11.6% (10) patients were diagnosed with community-acquired CDI.

Regarding the antibiotic treatment, the percentages of patients who received multiple antibiotics during hospitalization was higher in Group A (30% versus 22%). A statistical significance was found in both groups regarding specific antibiotic treatment. Coinfection was strongly associated with the administration of carbapenems (*p* = 0.005), while mono-infection was associated with cephalosporins treatment (*p* = 0.05) (Table 4). CDI was treated mostly with vancomycin 69 (80.2%), 20 (23.2%) with metronidazole, and 3 (3.4%) with both antibiotics.

The association between CDI and COVID-19 had a significant impact on the outcome of the patients, prolonging the hospitalization length. As can be observed in Table 5, hospitalizations were significantly longer and oxygen therapy requirements were significantly higher in Group A (*p* = 0.0036; *p* = 0.001), but in terms of patients outcomes those with only COVID-19 had a higher mortality rate (6.9% vs. 3.4%).

When analyzing the ROC curves and the areas under the curve (AUCs), we identified that dyspnea (AUC = 0.677; IC95%: 0.489–0.866; *p* = 0.083) and asthenia (AUC = 0.665; IC95%: 0.502–0.828; *p* = 0.106) had the strongest correlation with mortality (Figure 2, Table 6).

Regarding the laboratory findings, alanine aminotransferase (ALT) over 48 U/L was the variable with the strongest predictive rate for a fatal outcome, with 56% sensitivity and 39% specificity (AUC = 0.602; IC95%: 0.489–0.866; *p* = 0.083) (Figure 3, Table 7).

Survival analyses were conducted with KaplanMeier curves, which revealed that, in coinfected male patients, the probability of survival decreased to 70% in the first 21 days of illness. Similarly, the probability of survival decreased to 80% in the first 25 days of illness for coinfected patients from rural areas.

According to Table 8, the risk factors associated with mortality in Group A were male gender (*p =* 0.017) and older age (*p* = 0.001).

## 3. Discussion

The respiratory tissue is widely recognized as a primary target of SARS-CoV-2 infection, however, it is important to acknowledge that the detrimental effects of this virus are not confined solely to the pulmonary system. The virus can also attack tissues made of ACE2 and TMPRSS2-expressing cells, which are part of the gastrointestinal tract, heart, or kidneys [21]. Therefore, within the context of the COVID-19 pandemic, healthcare professionals must prioritize comprehensive diagnostic evaluations that encompass not only COVID-19, but also other potential pathologies, including gastrointestinal diseases, to ensure precise and accurate differential diagnoses. For CDI, the main risk factors requiring attention are antibiotic use, age over 65, immunodeficiency, and inflammatory bowel diseases [7]. In this study, we have shown that an unwanted consequence of antibiotic use is the occurrence of CDI, which comes as an accomplice, to the evolution of viral pathology (COVID-19). Antibiotics are life-saving drugs, having a vital role in infection management and they should be used when benefits are expected to outweigh the risks. The concern for microbiome imbalances in CDI is justified, but should not prevent a patient from receiving proper treatment with antibiotics when it is needed. Thus, it is important to know the optimal time and dose for their administration [22], in order to limit the intestinal imbalances associated and the antibiotic resistance rates, which are already alarmingly high at a local level [23,24,25,26]. Empiric antibiotic treatment in cases of SARS-CoV-2 infection is usually given when community-acquired bacterial pneumonia is suspected [27,28].

Overall, the number of coinfected patients in our study represented only 1.66% of all patients with SARS-CoV-2 infections admitted in the same hospital during the analyzed period. Their mean age was 61.96 years (ranging from 24 to 88), of which almost half were over 65. In most studies, age over 65 years was one of the main risk factors for this infection. For example, in a study from Poland, which investigated the risk factors for CDI, 53% of patients were over 65 years of age [29]. A report from the Center for Disease Control and Prevention mentioned that CDIs occur more frequently among patients aged 50 years or older, and especially over 65 years of age [30].

Based on our observations, it was common for patients with COVID-19 to present at the hospital approximately one week after the onset of symptoms, at which point the severity of the illness was notably elevated. As a result, suitable treatment was generally initiated with substantial delay. In addition to late presentation, coinfection with CDI led to further extension of the hospitalization length: coinfected patients spent from 7 to 83 days in hospital (17.3 days in average), compared to mono-infected patients, for whom the mean hospitalization period was 11.5 days. In a meta-analysis by Marra et al., the mean difference in length of stay between patients with and those without CDI varied from 3.0 to 21.6 days [31].

Further, 81.25% of patients presented unformed stools after admission, with an average number of 4.8 watery stools per day and a maximum of 21 stools per day. Li et al., in a tertiary care center in China, found that patients presented between 4 and 10 stools per day, while a systematic review conducted by Lambert et al. reported five or more stools per day in as many as 90% of hospitalized patients [32,33]. In our study, the mean number of stools (4.8) was relatively small, which led to a decreased level of dehydration among them. Hence, this factor is likely to have contributed to better evolution and outcomes. Contrary to the anticipated watery consistency typically reported in most studies [34,35], a significant majority of the patients included in our study (96.51%) described their stools as soft rather than watery.

During the pandemic, the usefulness of monitoring oxygen saturation was proven. An important predictor of in-hospital mortality for patients with COVID-19 was OS < 90% on admission (e.g., one study reported 28.46% of patients being admitted with OS below 80%) and, in addition, male sex has been correlated with high in-hospital mortality [36]. In our study, the incidence of OS levels under 90% on admission was relatively low (20.9%), with a higher incidence of decreased OS among men (median OS of 91.9% vs. 94.24% in women).

For CDI diagnosis, we tested for toxins A and B, as well as glutamate dehydrogenase (GDH) Antigen. Toxins A and B were detected in most cases, 80 (93%) toxin A, 72 (83%) toxin B, and 70 (81%) both toxins), while GDH was identified in all patients. This is a significant finding because toxin A can cause increased intestinal permeability and fluid secretion, while toxin B can determine severe colon inflammation [37]. Studies showed that toxin B participates in CDI severity, due to its involvement in both local and systemic host damage, and activation of the host inflammatory response [38,39,40]. Other studies found that toxin A is secreted more frequently than toxin B, consistent with our findings, but to a lesser extent [41,42].

Regarding inflammatory status, studies have shown increased proinflammatory markers in the serum of COVID-19 patients [43,44,45,46]. In CDI, CRP levels can predict severity [47,48]. Moreover, Guido et al. reported increased inflammatory markers levels in patients with both CDI and COVID-19 compared to a Group B [49]. In our study, the inflammatory markers were similarly elevated in both coinfected and mono-infected patient groups. CRP levels over 100 mg/L were associated with hospitalization periods of minimum 8 days and a mean of 17.2, and levels higher than 200 mg/L were associated with substantially longer hospitalization periods of up to 82 days and a mean of 24.4 days. Furthermore, higher CRP levels were correlated with oxygen desaturation and the need for ICU admission. These results highlight the role of assessing inflammation markers in the context of monitoring the progression of infectious diseases and, therefore, in deciding the optimal timing for initiating early antibiotic intervention.

Another aim of our study was to analyze the severity of COVID-19 disease in patients coinfected with CDI. While 53% of the patients included in our study experienced mild or moderate forms defined as SpO_2_ > 94% on room air, other studies reported higher rates reaching up to 81% [50,51]. Further, pneumonia was encountered in 52% of our patients, of which more than half featured ground-glass opacities on CT examination. In a study evaluating the relationship between chest CT imaging and COVID-19 pneumonia, the authors reported typical imaging features such as ground-glass opacities or mixed ground-glass opacities and consolidation for confirmed COVID-19 pneumonia cases that can help in the early screening and tracking of the disease [52]. The superior sensitivity of chest CT imaging in diagnosing COVID-19 was also demonstrated in two other studies (97% compared to 71% sensitivity of the RT-PCR method) [53,54]. Although such research encourages the use of CT imaging, this type of investigation is specifically recommended for detecting possible complications and alternative diagnoses rather than as an initial diagnostic approach, especially in less resourced countries and/or when the demand for CT scans is particularly high [55]. According to our multivariate analysis and regression modelling, dyspnea, fever, chest pain, as well as DM, obesity, and cardiac diseases were all significant predictors for the severity of ground-glass opacity in pneumonia (*p* = 0.01).

Regarding the relation between COVID-19 and CDI infection, it is important to carefully evaluate the data relating to the appropriate utilization of antibiotic therapy. Currently, resistance to antimicrobials is estimated to cause approximately 700,000 deaths worldwide annually [56]. Even if SARS-CoV-2 is a viral infection, the overuse of antibiotics in our hospital and elsewhere during the pandemic is alarming [57,58]. Patients with COVID-19 admitted in our hospital received mainly cephalosporins (55%) and carbapenems (24%), and records show that these two agents were given mostly in severe forms of COVID-19.

It is estimated that antibiotics are used in the treatment of more than 70% of COVID-19 cases, with respiratory fluroquinolones and cephalosporines most frequently given to patients coinfected with CD [59,60,61]. In our hospital, respiratory fluroquinolones were rarely used (1%), while the administration of cephalosporines was common practice (66%), especially ceftriaxone given to patients under 65 years, followed by carbapenems 40% (imipenem, meropenem) [62,63,64,65].

Comparing Group A versus Group B, we noticed that latter required significantly increased oxygen therapy (*p* = 0.01), and longer hospitalization (*p* = 0.01). Moreover, in terms of symptomatology, coinfected presented significantly more loose stools, abdominal pain, fever, asthenia, myalgia and arthralgia, headache, and ageusia. Paraclinical investigations revealed that leucocytosis, and serum urea and IL-6 were significantly higher in the presence of coinfection with CD. Broad-spectrum antibiotics were administered extensively, with a preference for carbapenems in Group A and cephalosporins in Group B.

Last but not least, according to our results, the classic symptomatology (e.g., dyspnea, asthenia) was a significant predictor of mortality, as well as the ALT levels. In other studies, the presence of such symptoms was also correlated with a poor prognosis [66,67,68]. Additionally, our analysis of outcomes, including survival and mortality, revealed that the probability of survival decreased to 70% in the first 21 days of illness in the case of male patients, and to 80% in the first 25 days of illness in the case of patients from rural areas.

Regarding risk factors association with mortality concerning patients with COVID-19 and CDI, we observed that male gender (*p* = 0.017) and advanced age (*p* = 0.001) were associated with increased risk of mortality. Research investigating the mortality risk of COVID-19 among male patients has consistently found a significantly higher risk of mortality compared to females [69,70,71,72,73]. Moreover, studies from the United States showed a higher mortality rate in males as compared to females in terms of only CDI [74,75], but considering patients with the association of COVID-19 and CDI, to the best of our knowledge, we found no data indicating a correlation between gender and its impact on the mortality rate. Notwithstanding, according to our findings, a study has shown that advanced age was also a risk factor for mortality in patients with CDI and COVID-19 [76].

For at-risk patients, a dynamic risk assessment and supervision are required to prevent recurrent infection, an aspect that can further burden the already frail patients, thus significantly impacting the healthcare systems costs [77].

Before we conclude, it is worth noting that our study was subject to limitations such as the small number of included patients and the single-center design. Further, certain resources were scarce, such as the testing kits for various markers.

## 4. Materials and Methods

We conducted a retrospective, observational study using hospital data of patients admitted for COVID-19 during March 2020—January 2021 in “Sf. Parascheva” Infectious Diseases Clinical Hospital Iași, in North Eastern Romania. The study was formally approved by the institution’s research ethics committee and patients provided written consent on admission for their anonymized data to be subsequently used for scientific research purposes.

The inclusion criteria were adult age (>18 years) and a SARS-CoV-2 positive RT-PCR test. Patients with COVID-19 who exhibited gastrointestinal symptoms either upon admission or during their hospital stay were tested for CD toxins A and B (by chromatographic immunoassay qualitative testings), as well as GDH antigen. At the time, the indications for SARS-CoV-2 testing were respiratory symptoms (cough, fever, dyspnea, dysphagia), and for CDI—digestive symptoms (abdominal pain, diarrhea, vomiting). We included patients with COVID-19 who presented digestive symptoms before admission and patients who developed CDI during hospitalization. Patients were excluded if they were under 18 years of age. Hospital admission for COVID-19 was considered the baseline.

The database was compiled and processed in SPSS (IBM Corp. Released 2021. IBM SPSS Statistics for Windows, Version 28.0. Armonk, NY, USA: IBM Corp). Apart from descriptive statistical analysis, univariate tests were performed to compare results and obtain the corresponding *p* values. Statistical significance was defined as a *p* value of less than 0.05 s. There was no adjustment for multiple comparisons. We assessed predictive value by calculating the areas under the receiver operating characteristic (ROC) curves (AUCs).

## 5. Conclusions

The diagnosis of SARS-CoV-2 infection and CD coinfection can be challenging due to the presence of similar gastrointestinal symptoms in both conditions. According to our results, the main risk factors for severe forms of COVID-19 were smoking and the association of DM, on one hand, as well as the preventable use of antibiotics, on the other. A patient with COVID-19 should not be exposed to antibiotics unnecessarily, especially during the early phase, as consequences may be severe, including CDI. In our study, in-hospital CD coinfection occurred mostly when carbapenems were administered and, importantly, it caused significant inflammatory syndrome and prolonged hospitalization, but in terms of deaths we did not find a significant impact. Nevertheless, risk factors associated with mortality in patients with COVID-19 and CDI were male gender and advanced age. Our results also suggest that patients admitted for COVID-19 who become infected with CD are vulnerable to worse outcomes. Consequently, CDI occurrence in a patient with COVID-19 reflects how our excessive use of antibiotics is undermining our best efforts to treat new infectious conditions effectively. Winning the fight against antibiotic resistance is of paramount importance to the future of human health, and it requires more judicious, evidence-based antibiotic therapy practices.

## Figures and Tables

**Figure 1 antibiotics-12-01091-f001:**
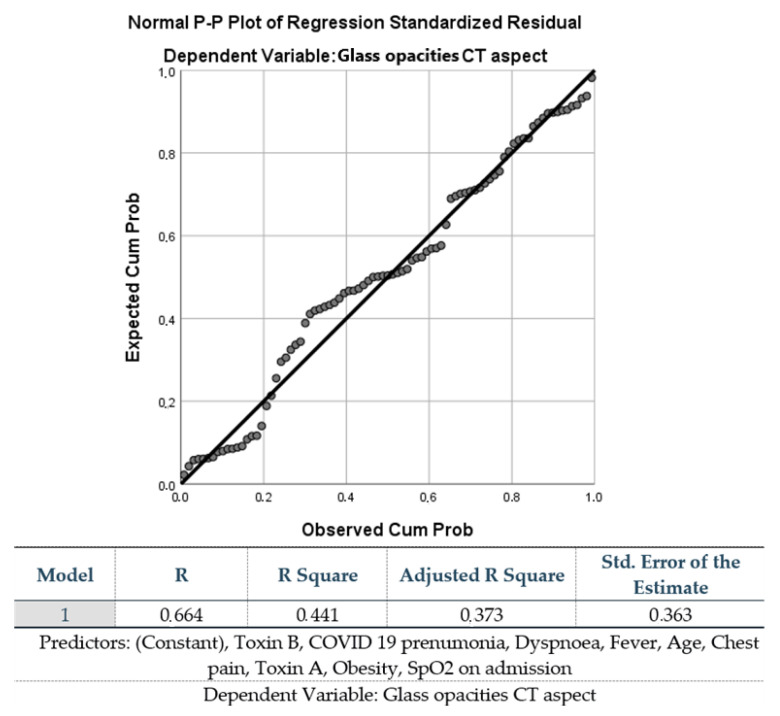
Predictors associated with glass opacities seen on the CT-scans of patients’ lungs.

**Figure 2 antibiotics-12-01091-f002:**
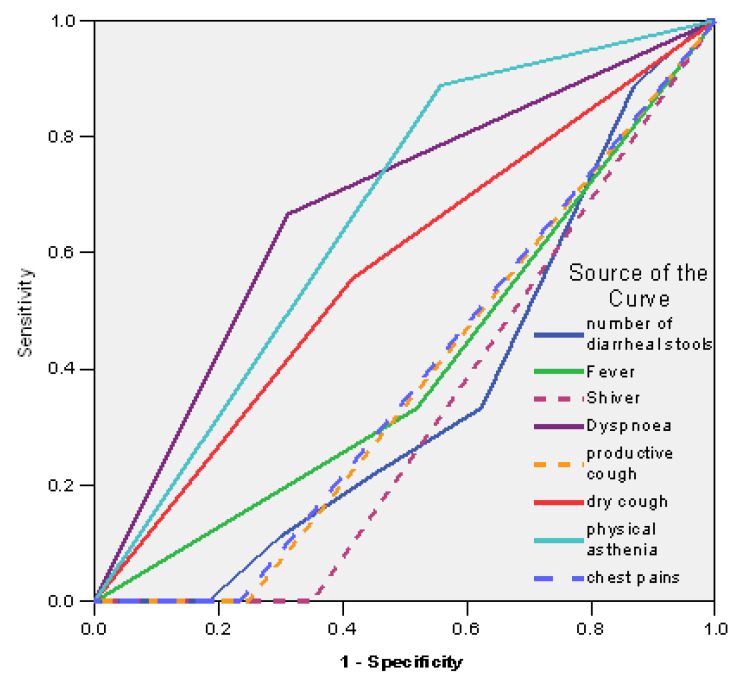
ROC curves for signs and symptoms associated with mortality.

**Figure 3 antibiotics-12-01091-f003:**
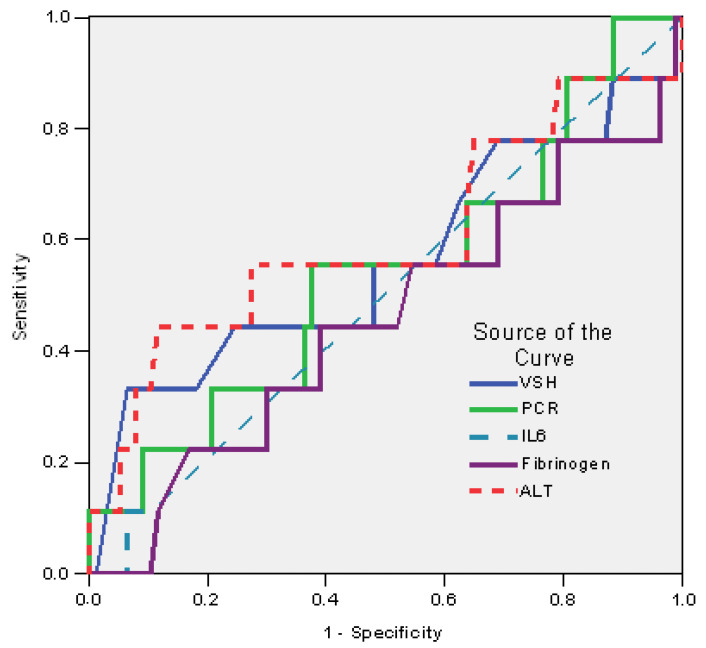
ROC curves for laboratory findings predicting mortality.

**Table 1 antibiotics-12-01091-t001:** Symptoms comparison in Group A (*N =* 86) depending on COVID-19 severity.

Symptomatology	COVID-19 Severe Form of Disease (*N =* 36)	COVID-19 Mild/Moderate Form of Disease (*N =* 50)	*T*-Value	*p*-Value
Fever	19 (52.7%)	26 (52%)	0.70	0.944
Shivers	13 (36.1%)	15 (30%)	0.591	0.550
Productive cough	8 (22.2%)	11 (22%)	0.024	0.981
Dry cough	20 (55.5%)	17 (34%)	2.016	0.047
Asthenia	22 (61.1%)	27 (54%)	0.651	0.517
Myalgia/arthralgia	10 (27.7%)	9 (18%)	1.046	0.299
Chest pain	8 (22.2%)	11 (22%)	0.024	0.981
Odynophagia	2 (5.5%)	6 (12%)	83.98	0.289
Headache	9 (25%)	14 (28%)	84	0.760
Anosmia	3 (8.3%)	6 (12%)	84	0.589
Ageusia	2 (5.5%)	6 (12%)	83.98	0.289

**Table 2 antibiotics-12-01091-t002:** Comparison of laboratory findings in Group A depending on COVID-19 severity in (*N =* 86).

	Mean of Severe Form of COVID-19 (*N =* 36)	Mean of Non Severe Form of COVID-19 (*N =* 50)	*t*-Value	*p*-Value
WBC (cells/µL)	10796.67	8253.20	1.914	0.06
Neutrophils (%)	77.55	67.52	2.722	0.008
Lymphocytes (%)	13.95	23.06	−3.074	0.03
Thrombocytes (10^3^ cells/µL)	288.79	256.10	1.085	0.2
LDH (UI/L)	369.89	274.33	2.483	0.01
D-dimer (FEU µg/mL)	143.45	151.56	−0.065	0.9
Interleukine-6 (pg/mL)	25.69	43.22	−0.76	0.5
Ferritin (ng/mL)	939.76	804.42	0.37	0.7
CRP on admission (mg/dL)	98.87	71.11	1.979	0.05
CRP at discharge (mg/dL)	46.58	55.14	−0.477	0.6
ESR (mm/h)	80.00	60.73	2.437	0.01
Fibrinogen (mg/dL)	9.01	7.36	0.319	0.7
INR	1.16	1.24	−0.58	0.5
Glucose (mg/dL)	140.92	118.09	1.992	0.05
Creatinine (mg/dL)	1.21	1.39	−0.6	0.5
ALT (U/L)	52.72	53.16	−0.042	0.9

Abbreviations: WBC—white blood cells count; LDH—lactate dehydrogenase; CRP—C-reactive protein; ESR—erythrocyte sedimentation rate; INR—International normalized ratio; ALT—alanine aminotransferase. Note: CRP levels were measured on admission as well as at discharge.

**Table 3 antibiotics-12-01091-t003:** The treatment of COVID-19 in Group A (*N =* 86).

COVID-19 Treatment	Mild/Moderate COVID-19 *N =* 50	Severe COVID-19 *N =* 36	*p*-Value	OR	CI 95%
Cephalosporins (3rd generation)	37 (74%)	20 (55.6%)	0.009	5.5914	1.4926–20.9459
Aminopenicillins	3 (6%)	3 (8.3%)	-	-	-
Macrolides	6 (12%)	2 (5.6%)	0.277	3.15	0.6176–16.0669
Carbapenems	14 (28%)	21 (58.3%)	0.04	2.45	1.0848–5.5335
Linezolid	3 (6%)	4 (11.1%)	-	-	-
Favipiravir	2 (4%)	0	-	-	-
Lopinavirum + ritonavirum	27 (54%)	12 (33.3%)	0.079	0.4259	0.1752–1.0357

Abbreviations: OR—odds ratio; CI—confidence interval.

**Table 4 antibiotics-12-01091-t004:** Antibiotic treatment during hospitalization.

Antibiotic Treatment	GROUP A *N* = 86	GROUP B *N* = 86	Total *N* = 172	*p*-Value	OR	CI 95%
Cephalosporins (3rd gen.)	57 (66.2%)	64 (74.4%)	121 (70.3%)	0.05	0.5887	0.3287–0.9621
Aminopenicillins	6 (6.9%)	10 (11.6%)	16 (9.3%)	0.299	0.5182	0.1816–1.4789
Macrolides	8 (9.3%)	7 (8.1%)	15 (8.7%)	0.790	1.2842	0.4491–3.6723
Carbapenems	35 (40.6%)	15 (17.4%)	50 (35.3%)	0.005	2.5926	1.3197–5.0933
Linezolid	7 (8.1%)	2 (2.3%)	9 (5.2%)	0.088	0.2456	0.0499–1.2099
Fluoroquinolones	1 (1.1%)	6 (6.9%)	7 (4%)	0.055	0.1435	0.017–1.2122
Association of antibiotics	26 (30%)	19 (22%)	45 (52.3%)	0.504	1.3076	0.675–2.5332

Abbreviations: OR—odds ratio; CI—confidence interval.

**Table 5 antibiotics-12-01091-t005:** Patients outcome in COVID-19 patients compared to coinfected patients.

Parameters	GROUP A (*N =* 86)	GROUP B (*N =* 86)	t-Value	*p*-Value
Length of hospital stay (days) (mean)	17.31	11.56	−2.9467	0.0036
Transfer to ICU	3 (3.4%)	10 (11.6%)	−1.581	0.117
Fatalities	3 (3.4%)	6 (6.9%)	−3.484	0.496
Oxygen therapy	54 (62.7%)	28 (32.5%)	4.246	0.001

Abbreviations: ICU—intensive care unit.

**Table 6 antibiotics-12-01091-t006:** AUC values for signs and symptoms predicting mortality.

Test Result Variables	AUC	Std. Error (a)	Asymptotic Sig. (b)	Asymptotic 95% Confidence Interval
No diarrhea	0.354	0.081	0.152	0.194–0.513
Fever	0.407	0.098	0.363	0.214–0.599
Shivers	0.325	0.074	0.087	0.180–0.469
Dyspnea	0.677	0.096	0.083	0.489–0.866
Productive cough	0.377	0.082	0.228	0.215–0.538
Dry cough	0.570	0.102	0.494	0.371–0.769
Asthenia	0.665	0.083	0.106	0.502–0.828
Chest pains	0.383	0.083	0.253	0.220–0.547

Abbreviations: AUC—Area Under the Curve; Note: The test result variables: number of diarrheal stools, fever, shiver, dyspnea, productive cough, dry cough, physical asthenia, and chest pains have at least one tie between the positive actual state group and the negative actual state group. Statistics may be biased; (a) Under the nonparametric assumption; (b) Null hypothesis: true area = 0.5.

**Table 7 antibiotics-12-01091-t007:** AUC values for laboratory findings predicting mortality.

Test Result Variable(s)	AUC	Std. Error (a)	Asymptotic Sig. (b)	Asymptotic 95% Confidence Interval
ESR	0.563	0.120	0.539	0.328–0.797
CRP	0.541	0.108	0.688	0.330–0.752
IL-6	0.502	0.102	0.983	0.302–0.702
Fibrinogen	0.455	0.109	0.662	0.241–0.670
ALT	0.602	0.120	0.317	0.367–0.837

Abbreviations: ESR—erythrocyte sedimentation rate, CRP—C-reactive protein; IL-6—interleukin 6, ALT—alanine aminotransferase, AUC—Area Under the Curve; The test result variable(s): ESR, IL-6, Fibrinogen, ALT have at least one tie between the positive actual state group and the negative actual state group. Statistics may be biased; (a) Under the nonparametric assumption; (b) Null hypothesis: true area = 0.5.

**Table 8 antibiotics-12-01091-t008:** Risk factors associated with mortality among patients from Group A.

Variable	Favourable Outcome Patients *N =* 83	Deceased Patients *N =* 3	Fisher’s Test *p*-Value
Male gender	42 (50.6%)	3 (100%)	0.017
Urban area of residence	54 (65%)	1 (33.3%)	0.093
Mean age	60.8	77.33	0.001
Associated pathologies:			
DM	20 (24%)	0 (0%)	0.093
CV	48 (57.8%)	0 (0%)	0.078
Neurological	14 (16.8%)	1 (33.3%)	0.125
Respiratory	3 (3.6%)	0 (0%)	0.109
Oncological	9 (10.8%)	2 (66.6%)	0.303
Previous antibiotic treatment	25 (30.1%)	1 (33.3%)	0.269
Hospitalization length (mean)	17.36	16	0.275
ICU admission	3 (3.6%)	0 (0%)	0.227
Severe COVID-19 form	35 (42.1%)	2 (66.6%)	0.230

Abbreviations: DM—diabetes mellitus; CV—cardio-vascular; ICU—intensive care unit.

## Data Availability

Not applicable.

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
