# Peer review of "Symptoms, Treatment, and Outcomes of COVID-19 Patients Coinfected with Clostridioides difficile: Single-Center Study from NE Romania during the COVID-19 Pandemic"

_antibiotics, 2023, doi:10.3390/antibiotics12071091_

Round 1
Reviewer 1 Report
The article, titled: Symptoms, Treatment, and Outcomes of COVID-19 Patients 2 Coinfected with Clostridioides difficile: Single-center Study from NE Romania during the COVID-19 Pandemic deals with a detailed analyses on the clinical signs and symptoms, the laboratory parameters and mortality of differences between a group of Covid-19-patients coinfected with Clostridium difficile (86) and a control group (86) having Covid-19 infection only. The results are demonstrated on different diagrams also.
This lengthy article includes only 1,66% of the total number of patients admitted to the hospital in the study period. It is not clear enough that how many tests were necessary to yield 86 C. difficile positive results. Their results do not contain any novelty, the longer hospital stay and “poorer outcomes”, is known from the literature, together with the susceptibility factors as diabetes, old age, any other underlying diseases, but in the table 6, the number of Covid-19 + C. difficile infected patients transferred to ICU was 3,4%, (4 patients) and with no mortality, but in the control group (Covid-19 infection only), the ICU transfer was 11,6% (13-14 patients) and the mortality rate was 6.9% (8 patients).
These contradictions should be addressed and redundancies eliminated, like the higher mortality of males, which is mentioned several times. After appropriate corrections, the article might be published.
Reviewer 2 Report
1. The challenges posed by this pathology are still remarkable. Line 41 change it to pathogen
2. line 316 methodology ,authors write “Further, in order to divide the patients into two groups, depending on the association 316 of CDI, we tested the patients for CD toxins A and B (by chromatographic immunoassay 317 qualitative testing’s) as well as GDH antigen”. It is not clear whether the tests were done earlier or prospectively done on stored sample as authors claim it be retrospective data Also it was an observational study or analytical??
3. Line373equal number of patients with COVID-19 as the control group. Authors are describing covid infected ( as mod & severe category) with or without CDI. Why is the non-CDI group labelled as control group?
4. Initially authors write that some patients were presenting with diarrhoae, so were those patients already on antibiotics . Its not clear whether antibiotic taken was recorded during their hospitalization only?
5. In Table 3 when were the patients s administered antibiotics which day of hospitalization.
6. Line 116:2.2. Analysis of clinical and paraclinical features of CDI in coinfected COVID-19 patients 116 The results obtained in terms of symptomatology in mono-infected and coinfected 117 patients were significantly different: dyspnea (50% vs 32%), asthenia (81,39% vs. 57%), 118 anosmia (21% vs. 10%), ageusia (19,76% vs. 9%). In this statement was the severity of covid different in CDI coinfected vs not ?Also the parameters were worsening in monoinfected versus coinfected ? Does not sound convincing
7. Authors to clarify if it is retrospective data then were all patients at the time of admission were subjected to C difficle toxin assay.If yes what was the indication and what antibiotics had the patients consumed before admission. Or is it that C difficle can be community acquired?? .not clear
8. Line 123Table 4. Laboratory findings on admission for all the patients included in the study (coinfected and 123 mono-infected): the group should be marked A & B ( as mono or coinfected ) .otherwise its not clear which coloumn is representing the group?
9. Table 4 can be deleted as only one parameter WBC is different which can be highlighted in the text.
10. Line141Table 6. Patients outcome in COVID-19 patients compared to coinfected patients. Length of stay is not mentioned whether in days ??
11. Fig 4& 5 not required
12. Table 9 : authors to provide explanation
13. Line 191 in Discussion resistance rates, which are 190 already alarmingly high at a local level. U. authors should give reference for this local burden
14. Discussion 202:when was sampling done for CDI ? Not clear Since all patients were admitted during pandemic ,authors should clarify whether CDI testing became mandatory or was it a part of this study?. Were any other bacterial or fungal coinfection detected??
15. How many of Coinfected CDI belonged to severe category of CoV 19 ? also how many had received carbapenems in coinfected group? severe covid were only 36 versus 50 in mod group.
16. Overall the data is confusing. The authors can just restrict to CDI infected in covid patients vs not infected covid patient and analyse the Data. The mortality cannot be associated to CDI as OR was not significant
1. The challenges posed by this pathology are still remarkable. Line 41 change it to pathogen
2. line 316 methodology ,authors write “Further, in order to divide the patients into two groups, depending on the association 316 of CDI, we tested the patients for CD toxins A and B (by chromatographic immunoassay 317 qualitative testing’s) as well as GDH antigen”. It is not clear whether the tests were done earlier or prospectively done on stored sample as authors claim it be retrospective data Also it was an observational study or analytical??
3. Line373equal number of patients with COVID-19 as the control group. Authors are describing covid infected ( as mod & severe category) with or without CDI. Why is the non-CDI group labelled as control group?
4. Initially authors write that some patients were presenting with diarrhoae, so were those patients already on antibiotics . Its not clear whether antibiotic taken was recorded during their hospitalization only?
5. In Table 3 when were the patients s administered antibiotics which day of hospitalization.
6. Line 116:2.2. Analysis of clinical and paraclinical features of CDI in coinfected COVID-19 patients 116 The results obtained in terms of symptomatology in mono-infected and coinfected 117 patients were significantly different: dyspnea (50% vs 32%), asthenia (81,39% vs. 57%), 118 anosmia (21% vs. 10%), ageusia (19,76% vs. 9%). In this statement was the severity of covid different in CDI coinfected vs not ?Also the parameters were worsening in monoinfected versus coinfected ? Does not sound convincing
7. Authors to clarify if it is retrospective data then were all patients at the time of admission were subjected to C difficle toxin assay.If yes what was the indication and what antibiotics had the patients consumed before admission. Or is it that C difficle can be community acquired?? .not clear
8. Line 123Table 4. Laboratory findings on admission for all the patients included in the study (coinfected and 123 mono-infected): the group should be marked A & B ( as mono or coinfected ) .otherwise its not clear which coloumn is representing the group?
9. Table 4 can be deleted as only one parameter WBC is different which can be highlighted in the text.
10. Line141Table 6. Patients outcome in COVID-19 patients compared to coinfected patients. Length of stay is not mentioned whether in days ??
11. Fig 4& 5 not required
12. Table 9 : authors to provide explanation
13. Line 191 in Discussion resistance rates, which are 190 already alarmingly high at a local level. U. authors should give reference for this local burden
14. Discussion 202:when was sampling done for CDI ? Not clear Since all patients were admitted during pandemic ,authors should clarify whether CDI testing became mandatory or was it a part of this study?. Were any other bacterial or fungal coinfection detected??
15. How many of Coinfected CDI belonged to severe category of CoV 19 ? also how many had received carbapenems in coinfected group? severe covid were only 36 versus 50 in mod group.
16. Overall the data is confusing. The authors can just restrict to CDI infected in covid patients vs not infected covid patient and analyse the Data. The mortality cannot be associated to CDI as OR was not significant
Reviewer 3 Report
Dear authors,
The manuscript with the title “Symptoms, Treatment, and Outcomes of COVID-19 Patients 2 Coinfected with Clostridioides difficile: Single-center Study 3 from NE Romania during the COVID-19 Pandemic” describes clinical course of covid-19 and covid-19 an CDI coinfection in patients Romania.
in the introduction, the authors present commonly known facts about covid-19 and separately facts about CDI, but the reader does not learn anything about the possible influence on the course and outcome of covid-19 treatment, which is later discussed (not know previously). Does CDI interfere with covid-19 somehow? Reader also learns nothing about the problem of CD outbreaks in covid-19 wards during the covid-19 epidemic and the reasons for the increased occurrence of CD during this time.
In result section it is not it clear whether these 86 CD patients are patients who were admitted to the hospital due to CDI (and was covid-19 cofounded) or they acquired CDI in the hospital as a nosocomial infection during their covid-19 infection (due to broad spectrum antibiotics etc.). It is also not clear whether these 86 patients present an outbreak (or more outbreaks) on covid-19 wards.
This are two main reasons why it is not possible to compare the outcome of covid-19 disease in CDI coinfected patients with those not- coinfected- it:
-it is not clear if these 86 patients were primary admitted to the hospital due to covid-19 (later it is stated that only 52% had pneumonia) or if their main problem was CDI (and covid-19 was only asymptomatic bystander) or if they had severe covid-19 pneumonia and developed later CDI as a nosocomial infection). To my opinion the group of CDI patients is not homogenous enough.
- covid-19 changed dramatically during 2-year period due to different strain (omicron with different clinical course and less severe cases) so it is important to compare clinical course and severity of the diseases during the same covid-19 “period” , especially when comparing it to another much smaller group (>5000 patients vs 86 patients).
In this context are the tables difficult to understand:
-table 1: although it compares severe and mild disease in covid-19 -CDI patients there is nothing new in it (only symptoms and signs of severe disease are not included)- the symptoms of covid-19 are well known by now.
- Table 2- similar to table 1: covid-19 is well known by now
-table 3. It is interesting that no remdesivir was used
- Table 4- doesn’t make sense: comparing lab results of 2 different diseases (maybe 3: covid-19, CDI and covid-19 with CDI)
Etc.
The authors do not explain why is CDI coinfection to their opinion important for covid-19 outcome. I would recommend that authors instead of comparing the outcome of covid-19 in CDI patients to analyze the reasons for CDI infections maybe during different covid-19 waves:
- maybe the consumption of broad-spectrum antibiotics changed during 2-year period
- did this influenced the occurrence of CDI, were the wards overcrowded during different waves?
- Where they sporadic infections or outbreaks in covid-19 wards?
Best regards
